# The Synergy between Glutathione and Phenols—Phenolic Antioxidants Repair Glutathione: Closing the Virtuous Circle—A Theoretical Insight

**DOI:** 10.3390/antiox12051125

**Published:** 2023-05-19

**Authors:** Mirzam Carreon-Gonzalez, Juan Raúl Alvarez-Idaboy

**Affiliations:** Departamento de Física y Química Teórica, Facultad de Química, Universidad Nacional Autónoma de México, Mexico City 04510, Mexico; mirzam.carreon@quimica.unam.mx

**Keywords:** synergy, phenolic antioxidants, glutathione repair

## Abstract

Glutathione (GSH) and phenols are well-known antioxidants, and previous research has suggested that their combination can enhance antioxidant activity. In this study, we used Quantum Chemistry and computational kinetics to investigate how this synergy occurs and elucidate the underlying reaction mechanisms. Our results showed that phenolic antioxidants could repair GSH through sequential proton loss electron transfer (SPLET) in aqueous media, with rate constants ranging from 3.21 × 10^6^ M^−1^ s^−1^ for catechol to 6.65 × 10^8^ M^−1^ s^−1^ for piceatannol, and through proton-coupled electron transfer (PCET) in lipid media with rate constants ranging from 8.64 × 10^6^ M^−1^ s^−1^ for catechol to 5.53 × 10^7^ M^−1^ s^−1^ for piceatannol. Previously it was found that superoxide radical anion (O_2_^•−^) can repair phenols, thereby completing the synergistic circle. These findings shed light on the mechanism underlying the beneficial effects of combining GSH and phenols as antioxidants.

## 1. Introduction

Oxygen is a chemical element that is widely distributed on Earth, so it is undoubtedly essential for life. Despite being necessary for efficient energy production in mitochondria, it is also toxic to aerobic systems [1,2]. When oxygen is partially reduced, it gives rise to reactive oxygen species (ROS) and free radicals (R^•^), which can become dangerous products of cellular metabolism if produced in high concentrations. Although these species are necessary for the proper development of various biological functions [3,4,5,6], an imbalance between their production and the body’s ability to purify them gives rise to what is known as oxidative stress (OS) [7,8].

Although the body has enzymatic defense mechanisms to counteract the adverse effects caused by OS, a wide variety of endogenous and exogenous factors increase the production of ROS and R^•^, so this pathway may be insufficient to protect against permanent damage [9,10,11,12,13,14,15,16]. Fortunately, there are alternative ways to reduce the concentrations of these species in the body, such as a high intake of foods rich in antioxidants [17,18,19].

Antioxidants are chemical species capable of delaying, preventing, and even repairing the damage caused to other molecules because of OS [20]. However, since the oxidative process implies a large set of pathways, no antioxidant is entirely effective in combating the adverse effects produced by this process. For this reason, mixtures of antioxidants have been suggested since there is evidence of their synergistic effect [21,22].

The antioxidant activity of a species implies its oxidation, which is why it could partially or totally lose its functions if it is not repaired or regenerated by some other species in the medium. In this way, other antioxidants, which could not carry out the same action due to their nature, could repair them by another mechanism and thus generate a synergistic cycle [23].

The mechanisms through which antioxidants exert their properties depend on the nature, concentration, and the cellular medium in which they are found. Therefore, different pathways of action have been identified. The two main mechanisms of action are single electron transfer (SET) and formal hydrogen transfer (FHT). Thus, an antioxidant that acts via FHT can be regenerated via SET or vice versa [23].

The synergistic activity of antioxidants has been used in different areas, for example, in the food industry, the combination of antioxidants is studied to find a mixture that allows greater efficiency in food preservation, as well as a decrease in the number of synthetic additives [24,25,26,27]. In the pharmaceutical industry it has been found that the use of a mixture of antioxidants with some drugs increases therapeutic efficacy and decreases cell damage [28,29,30,31].

Chemical species are frequently classified as oxidants or antioxidants based on the predominant function they exhibit; it has been found that several of these species could exhibit dual behaviour depending on physiological conditions [32,33,34,35]. There is evidence, for example, that polyphenols could act as pro-oxidants despite being known for their great antioxidant activity [36,37,38]. The preceding allows us to suppose that if an antioxidant can present a dual behaviour, an oxidant also can. One of the cases that has attracted a great deal attention is that of the O_2_^•−^, an abundant by-product of aerobic respiration, highly recognized for its oxidizing effects; can repair oxidized DNA through a SET mechanism [39,40,41,42]. It has been previously proposed that superoxide anion radical can repair phenolates in water with a pH = 7.4 [43,44,45]. Although there are experimental reports that show the synergy between O_2_^•−^ and phenols (melanins) [46] or other radicals such as nitroxides [47], the reaction conditions, as well as the mechanisms proposed in these reports, differ from what is studied in the present work. However, they provide evidence of the importance and application of synergy with O_2_^•−^.

In this context, the possibility arises that a thiol-type antioxidant may be regenerated by phenolic-type antioxidants, known to act via SPLET mechanism in aqueous media [48,49,50,51,52] and by their ability to be repaired by O_2_^•−^ under physiological conditions [43,44,45]. This way, a synergistic cycle would be generated, in which a thiol sacrifices itself to repair damaged biomolecules. Then, it is repaired by a phenol, which, in due time, is regenerated by the superoxide radical anion.

Thiol-type compounds have been studied as potential antioxidants since sulfur is an element embedded in a range of biologically important molecules. It performs a crucial role in the organism [53,54,55,56,57,58]. For example, cysteine is a non-essential sulfur-containing amino acid in humans but is essential for protein synthesis, detoxification, and diverse metabolic functions [59,60]. Tiopronin is a low molecular weight synthetic analogue of glutathione used to treat cystinuria, rheumatoid arthritis, mercury, copper poisoning, etc. [61]. However, the thiol-type antioxidant of greatest interest is glutathione. Glutathione is a tripeptide made of the amino acids: glutamate, cysteine, and glycine. Given its concentration in the cell, around (0.1–10) mM [62,63,64,65], it is considered one of the most abundant and essential endogenous antioxidants. Alongside many other physiological activities, as an antioxidant, it presents a unique ability to repair carbon centre radicals via a hydrogen atom transfer (HAT) mechanism with rate constants close to the diffusion-limited regime [66]. The low reactivity of phenolic antioxidants for this reaction and the inertness of carbon centre radical against electron donor antioxidants make GSH an essential molecule in fighting OS. Hence, it is important that it can be repaired after forming a thiyl radical and before recombination.

On the other hand, phenolic-type antioxidants are widely distributed in the human diet. Their intake has been shown to have health benefits, including free-radical scavenging, estrogenic, anti-inflammatory, and anticancer activity [36,38,67].

Since previous works have shown the ability of thiol-type antioxidants to repair molecules of biological interest [68] and the capability of phenols to regenerate under physiological conditions [43,44,45], the present work focuses on studying the repair of glutathione radical (GS^•^) by some phenolic antioxidants. They are catechol (Cat), pyrogallol (Pyr), trans-resveratrol (Res), and trans-piceatannol (Pic). This possibility would be crucial for living organisms since, at the end, O_2_^•−^ which is continuously produced by respiration, would create an endless (except for recombination) cycle. It should be noted that outside the living cells, the three components, GSH, phenol, and O_2_^•−^, never coexist unless a specific experiment is designed for this proposal.

Phenolic antioxidants were chosen based on their functionality, their easy intake, and their multiple biological activities. For example, catechol and pyrogallol were chosen as basic structures since they are present in a wide range of polyphenols, such as catecholamines, phenolic acids, catechins, flavonoids, stilbenes, etc. Additionally, catechol is used as an astringent, antiseptic, and as a precursor to drugs, pesticides, and cosmetic products [69,70,71]. It can be obtained naturally in small amounts, mainly through the juice or bark of the mimosa tree [69]. Pyrogallol is a natural product found in *Gunnera perpensa* and *Nigella glandulifera* [71,72,73,74]. This polyphenol has been shown to be effective against lung cancer [75]. Piceatanol and resveratrol are naturally occurring stilbenes found primarily in grapes, berries, peanuts, and red wine. Both species are acknowledged as antioxidants, and in addition to that, they offer a diverse range of health benefits, such as estrogenic, anti-inflammatory, and anticancer activities [76,77,78,79,80].

The polyphenols examined in this study are exogenous antioxidants that can be readily consumed through natural sources such as fruits and vegetables. Therefore, their bioavailability in the body varies depending on the individual’s diet, state of health, physiological conditions (such as pH and temperature), and the method of administration, among other factors [81,82,83,84]. For resveratrol, which is the phenol that presents more studies on its bioavailability, there is no consensus regarding the concentrations in which it is available in blood or tissues. According to several studies, the concentration of resveratrol in blood or tissues is low after a few hours of being consumed: of 25 mg administered orally, it was only possible to detect peak concentrations of <10 ng/mL in plasma, (0.5–2) h post-dose. Estimates of plasma concentrations of resveratrol plus total metabolites were considerably higher, around (400–500) ng/mL (≈2 mM) indicating a very low oral bioavailability for free resveratrol. However, studies in rats have shown that after prolonged administration (15 to 20 weeks), there is a saturation of the metabolism, which leads to an increase in resveratrol in plasma and, therefore in tissues [79]. Thus, if the intake of polyphenols is prolonged, their concentrations increase enough for their biological properties to be appreciable [77,79,82].

Although the synergy between thiolic and phenolic antioxidants has already been reported [24,85,86,87], it has not been studied under the same physiological conditions proposed in the present work. So, at pH = 7.4 in aqueous and lipid media, the reaction mechanisms through which these antioxidants enhance their effects are not well known. In Figure 1 and Figure 2, the complete proposed synergistic (virtuous) cycle is shown. Considering that the reactions between damaged DNA [66,88] and peptides [68,89,90] with thiols (show in blue) and between phenolates and superoxide [43,44,45] (show in green) has already been reported, the reactions between thiol radicals and phenols (show in black) are those studied in the present work, in aqueous and lipid medium, respectively.

At this point, it is important to mention that in reactions that involve electron transfer, the standard reduction potentials of the species involved play an essential role in the viability of the reactions. Standard reduction potentials indicate whether a species can accept electrons. The lower the value of potential, the better electron donor the species is. Therefore, the species is better antioxidant.

As far as we are aware, the standard reduction potential for all the species involved have not been reported through experimentation. However, there are studies on closely related systems that provide support for the proposed synergistic cycle. For glutathione and catechin, the standard reduction potentials vs. SHE at 37 °C are reported to be (0.310 ± 0.003) V and (0.281 ± 0.008) V, respectively [91]. This indicates that glutathione has a lower antioxidant activity compared to catechin. The same study shows that for a set of substituted catechins, the reduction potentials of these species are less than that of glutathione. These standard reduction potential values allow us to assume that the reaction between phenolate and glutathione radical (as shown in Figure 1) is thermodynamically favored.

Finally, given that under physiological conditions, the presence of molecular oxygen (O_2_) in the medium can lead to a reaction in competition with the reaction between alkyl radicals and glutathione. Although this part of the cycle is not the study objective in the present work, it is necessary to mention that since O_2_ reacts with alkyl radicals at constant rates close to diffusion, the first step shown in Figure 1 (initiation) might not occur. However, reactions of GSH with alkyl radicals are also diffusion controlled; therefore, the concentrations of the species involved play a determining role. In blood, the dissolved oxygen concentration is low (approximately 0.003 mL/100 mL plasma) [92,93] compared to the glutathione concentration (which can be up to three times higher), so the reaction between alkyl radicals and glutathione could be favored. This is discussed in detail in reference [66].

## 2. Materials and Methods

The main stationary points (reactants, transition states, and products) were computationally modeled for SET and FHT reaction pathways simulating physiological conditions at pH = 7.4 in aqueous and lipid media. Water and pentyl ethanoate were used as solvents to mimic the aqueous and lipid medium, respectively. Electronic calculations were performed with the Gaussian 09 software [94]. Geometry optimizations and frequency calculations were carried out at theory level M06-2X/6-311++G (d,p) [95] in conjunction with the continuum solvation model based on density (SMD) [96]. The number of imaginary frequencies allowed us to identify the local minima and the transition states (0 or 1, respectively). It was confirmed that the imaginary frequency corresponded to the expected movement along the reaction coordinate for the transition states (TS).

Intrinsic reaction coordinate (IRC) calculations confirmed that the transition states connected appropriately with the reactants and products. Relative energies were calculated with respect to the isolated reactants and corrected to 298.15 K. Solvent cage effects were included based on Okuno corrections considering the free volume theory [97]. The rate constants were calculated using a conventional transition state theory (TST) [98,99]:(1)kTST=σκkBThe−ΔG∘‡RT 
where k_B_ is the Boltzmann constant, h is the Planck constant, ΔG^°‡^ is the Gibbs free energy of activation, σ is the reaction path degeneracy, and κ represents the tunneling corrections, which were calculated using an asymmetric Eckart barrier [100]. For the reactions involving single-electronic transfers, the barriers of reaction were estimated using Marcus’ theory [101,102,103]:(2)$$\Delta G^{\circ ‡}=\frac{\lambda }{4}{\left(1+\frac{\Delta G_{RXN}^{\circ }}{\lambda }\right)}^{2}\;$$
where $\Delta G_{\mathrm{RXN}}^{\circ }$ is the free energy of reaction whose values were calculated according to Hess’s law. The reorganization energy, λ, were calculated according Equation (3) [104,105,106]:(3)$$\lambda =\lambda_{i}+\lambda_{0}\;$$
where λ_i_ is the internal energy and λ_0_ is the solvent reorganization energy. The values of λ_i_ were calculated as Equation (4) and the values of λ_0_ were calculated based on the 2-sphere model of Marcus according Equation (5) [104,105,106]:(4)$$\lambda_{i}=\Delta E-\Delta G_{RXN}^{\circ }\;$$
(5)$$\lambda_{0}=\Delta q^{2}\left(\frac{1}{\varepsilon_{\infty }}-\frac{1}{\varepsilon_{0}}\right)\left(\frac{1}{2r_{A}}+\frac{1}{2r_{D}}-\frac{1}{r_{AD}}\right)$$
where ΔE is the energy variation associated with the vibrational relaxation of the reactants after the vertical electron transfer occurred, Δq is the amount of charge transferred, r_A_, r_D,_ and r_AD_ are the effective radii of the acceptor (A), donor (D), and reactant complex (AD), and $\varepsilon_{\infty }$ and $\varepsilon_{0}$ are the static and optical dielectric constants of the solvent, respectively.

Rate constants that are close to, or within, the diffusion-limited regime was corrected by the Collins–Kimball theory [107] (Equation (6)) in conjunction with the steady-state Smoluchowski [108] (Equation (7)) and the Stokes–Einstein [109] approaches (Equation (9)):(6)$$k_{app}=\frac{k_{D}k^{TST}}{k_{D}+k^{TST}}\;$$
(7)$$k_{D}=4\pi RD_{AD}N_{A}\;$$
where R is the reaction distance, N_A_ is the Avogadro number and D_AD_ is the mutual diffusion coefficient of the reactants A and D and was calculated according to Equations (8) and (9) [110]:(8)$$D_{AB}=D_{A}+D_{D}\;$$
(9)$$D_{A\mathrm{or}D}=\frac{k_{B}T}{6\pi ŋa}\;$$
where T is the temperature, η is the viscosity of the solvent, and α is the radius of the solute. Total rate constants (k_total_) were corrected by the molar fractions at pH = 7.4. For this purpose, the experimental pka’s were used to calculate the molar fractions. They are 3.59 and 8.75 for glutathione [66], 9.24 for catechol [111], 8.94 for pyrogallol [112], 8.99 for resveratrol [113], and 7.86 for piceatannol [114]:(10)$$k_{total}=\alpha_{\left(DH\right)}\alpha_{\left(R^{•}\right)}k_{app}$$

Finally, the overall rate constants (k_overall_) were obtained as the sum of the k_total_ for both reaction mechanisms in the same solvent as implemented in the QM-ORSA protocol [115]:(11)$$k_{overall}=k_{total}^{SET}+k_{total}^{FHT}\;$$

## 3. Results and Discussion

### 3.1. Thermodynamics and Kinetics

Table 1 and Table 2 show the Gibbs reaction free energy values ($\Delta G_{\mathrm{RXN}}^{{}^{\circ}}$), the activation free energy (ΔG^°‡^) and the rate constants corresponding to the reactions between glutathione radical and phenols in aqueous and lipid mediums. The total rate constants (k_total_) were obtained through the conventional TST theory (according to Equation (1)) [98,99] and corrected for tunneling [100] and solvent liquid phase effects [97]. Reactions with rate constants close to the diffusion-limited regime were corrected by the Collins–Kimball theory (according to Equations (6)–(9)) [107,108,109,110]. Finally, considering that in the reaction medium and at pH = 7.4, the relative abundance of the species involved varies according to their pK_a_, the rate constants were corrected by their respective molar fraction (according to Equation (10)). Once the k_total_ was known, the global rate constants (k_overall_) were calculated as the sum of the k_total_ obtained for each reaction mechanism studied (according to Equation (11)).

The calculated values for the Gibbs free energy of reaction, the activation free energy, and the overall rate constants show that in an aqueous medium the SET mechanism is thermodynamically and kinetically favored. Since the phenolic antioxidant is partially deprotonated under physiological conditions, electron transfer occurs from phenolate to the GS^•^ radical, therefore, SPLET is the primary mechanism [116,117,118]. Even though the FHT mechanism is thermodynamically feasible, its activation free energy is considerably more prominent than in SET reactions, so, under physiological conditions, this mechanism does not contribute significantly. Therefore, phenolic antioxidants repair sulfur radicals by transferring an electron in aqueous media after losing a proton (See Table 1). Piceatannol is the best phenol-type antioxidant since by giving up an electron, its structure allows the radical that is generated to stabilize by resonance. Additionally, it can develop intramolecular hydrogen bonds in the transition state geometry (See Figure 1). Since some of the phenolic antioxidants used in the present work have more than one OH group in their structure, they can generate hydrogen bonds with the GS^•^ radical. Such interactions help stabilize the transition state geometry. Although it is known that the use of explicit solvent molecules can also help to stabilize the species involved in the reaction by hydrogen bonds, its use is not recommended in large systems (more than 20 atoms), especially if the reaction involves a transition state, since finding the right amount of explicit solvent molecules, their geometry, and the number of solvation spheres required to stabilize the system implies an enormous computational expense. The time invested in the study of how the explicit solvent molecules affect the stability of the system is not always directly proportional to the improvement of the results. For this reason, the SMD model [96] was used to simulate the reaction medium and no explicit solvent molecules were used.

As previously mentioned, for reactions involving electron transfer, the standard reduction potentials of the species involved can help predict whether such reactions will occur. Based on the experimental standard reduction potentials reported for glutathione and a group of catechins, it was suggested that the electron transfer from the phenolate to the GS^•^ radical would be thermodynamically favored. The free energy values of the reaction via SET allow us to verify that the initial approach is correct (See Table 1); that is, the phenolic antioxidants (in their anionic form) have the capacity to repair the GS^•^ radical by giving up an electron.

On the other hand, since a non-polar medium does not allow the stabilization of ionic species, the SET reactions do not occur in lipid medium. Hence, only the FHT mechanism is analyzed. In this medium, the FHT mechanism is favored since the molar fractions do not affect the rate constants (See Table 2).

As in the aqueous medium, piceatannol is an excellent antioxidant, given its ability to stabilize by resonance of the generated radical. Pyrogallol was found to be the best antioxidant in the lipid medium, probably because it is the species that could form most intramolecular hydrogen bonds in the transition state of all phenol-type antioxidants tested, and because the low BDE of the O-H bond.

The H-bond formation in the transition states deserves special attention. Depending on conformation, catechol and piceatannol may or may not be able to form a H-bond. We have calculated both transition states in both media for both reactions. The corresponding barriers in Gibbs energy differ by 1.27 kcal/mol and 1.67 kcal/mol for catechol and piceatannol in an aqueous medium (See Table 1), respectively, while in a lipid medium, they differ by 3.85 kcal/mol and 4.16 kcal/mol (See Table 2). This means that hydrogen-bonded transition states produce 3.93 times and 2.76 times larger rate constants for catechol and piceatannol in an aqueous medium (See Table 1), respectively, while in a lipid medium they are 664.62 times and 1506.81 times larger in comparison with the transition state without a H-bond (see Table 2). Since the human brain is nearly 60 percent fat and has a large concentration of catecholamines, this finding could be of great importance.

These results allow us to show that, in water, the H-bonds in the transition states are obviously much weaker since H-bond strength is inversely proportional to the dielectric constant of the medium as well as its ability to form H-bonds.

Figure 1 shows the geometries of H-bonded transition states in an aqueous and a lipid medium for catechol and piceatannol. For catechol, the H-bond distance in water is 1.84 Å and 1.71 Å in pentyl ethanoate, so the difference is 0.13 Å, which is quite large for an H-bond distance difference. For piceatannol, the H-bond distances are 1.85 Å and 1.75 Å, respectively, so the difference is 0.10 Å. Accordingly, the energy barriers for reactions involving transition states without H-bond are higher, consistent with the reaction between GS^•^ radical and resveratrol, the only phenol that cannot form a H-bond in the transition state geometry, being the species with the most significant barriers.

### 3.2. Mechanistic Analysis

A FHT mechanism is defined as one in which a hydrogen atom (H^•^) is transferred between the donor and the acceptor, in this case, between phenolic antioxidant and GS^•^ radical, respectively. Since a hydrogen atom is composed of a proton (H^+^) and an electron (e^−^), the transfer pathways of these particles can be concerted or sequential. HAT and PCET are the concerted type FHT mechanisms, while electron-proton sequential transfer (EPST) and proton-electron sequential transfer (PEST) are the sequential types. Although the same products are obtained from the HAT and PCET mechanisms, they differ in how the particles move: in the HAT mechanism, both particles move as a single species, that is, as a hydrogen atom (H^•^), while in the PCET mechanism, they move as independent particles (H^+^ and e^−^) [119,120]. To differentiate between the HAT and PCET mechanisms, two computational tools are used. The first is the analysis of the natural charge populations (NPA) and Hirshfeld spin densities on the donor, the acceptor, and the transferring nucleus as a function of the IRC. The second is the analysis of density surfaces of individually occupied molecular orbitals (SOMO) in the transition state geometry [120,121].

Figure 2 shows the expected behavior for a PCET mechanism [119,120,121,122,123] since the charge on the transferred nucleus is considerably larger (compared to the acceptor and donor), which is consistent with the transferred species H^+^. The charge crossing between the donor and acceptor in the lipid medium coincides with the fact that the net charge of the system is zero in this medium. However, such crossing does not occur in aqueous media since the system net charge is −1. Moreover, the spin density of the transferred nucleus is small (~0.5%), which is consistent with the transfer of a H^+^. The spin density crossing between the donor and acceptor indicates that that the species that make up the H^•^ are transferred in a single reaction step but as separate species, i.e., as a H^+^ and an e^−^. Since the charge and spin redistribution during the H^+^ transfer is abrupt, the slope in the crossing is pronounced which indicates that process is nonadiabatic [68,120]. As can be seen, the reaction medium does not affect the mechanism. Although the phenolic antioxidants tested in this work showed the ability to donate a hydrogen atom via PCET mechanism and thus repair the GS^•^ radical in an aqueous medium at pH = 7.4, these reactions are thermodynamically and kinetically unfavored with respect to the reactions of electron transfer (SET) (See Table 1). In such a way that the PCET mechanism does not contribute significantly to the antioxidant activity of the tested phenols; therefore, the SPLET mechanism is the main reaction mechanism through which phenolic antioxidants repair the GS^•^ radical in an aqueous medium at physiological pH.

Figure 3 shows the graphs of the NPA atomic charges and the Hirshfeld spin densities as a function of the IRC for the reaction between the glutathione radical and piceatannol. For this case, similar behavior can be observed, which means that the reaction mechanism is the same for the reactions between glutathione radical and phenol-type antioxidants, i.e., the reaction mechanism through which phenolic-type antioxidants repair the glutathione radical is PCET [119,120,121,122,123]. For the reaction in water, the slope in the crossing of spin densities is less abrupt, which is consistent with an adiabatic process [65,114].

The graphs of the atomic charges and spin densities as function of the IRC for the reactions between the glutathione radical and the rest of the phenol-type antioxidants are not shown since the mechanism is the same and the medium does not significantly affect the reaction mechanism.

Finally, Figure 4 shows the SOMO and HOMO orbitals at the transition state geometry. It is observed that the orbitals are delocalized over the donor and the acceptor, which confirms that the mechanism is PCET [119,120]. The SOMO and HOMO orbitals for piceatannol (See Appendix A) and for the rest of the phenol-type antioxidants, which are not shown, exhibit similar behavior. Therefore, it is ratified that the reaction mechanism is PCET. Appendix A show the cartesian coordinates of the transition states for the reactions between GS^•^ radical and piceatannol in both media.

## 4. Conclusions

The thermodynamic, kinetics, and mechanistic analysis make it possible to demonstrate the proposed synergistic cycle in which thiol-type antioxidants (glutathione), which sacrifices itself to repair molecules of biological interest, are regenerated via SPLET in the aqueous medium and via PCET in the lipid medium by phenolic antioxidants. As previously proposed, these are subsequently restored in the presence of superoxide in aqueous medium.

The large rate constants obtained, ranging from 3.21 × 10^6^ M^−1^ s^−1^ for catechol to 6.65 × 10^8^ M^−1^ s^−1^ for piceatannol in aqueous media and from 8.64 × 10^6^ M^−1^ s^−1^ for catechol to 5.53 × 10^7^ M^−1^ s^−1^ in lipid media, indicate that these reactions are expected to be highly competitive in biological systems.

## Data Availability

Data are contained within the article and Appendix A.

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
