# Peer review of "The Synergy between Glutathione and Phenols—Phenolic Antioxidants Repair Glutathione: Closing the Virtuous Circle—A Theoretical Insight"

_antioxidants, 2023, doi:10.3390/antiox12051125_

Round 1
Reviewer 1 Report
Please see attached file.

Author Response
We thank the Reviewer for the comments that certainly improve the manuscript.
Comments:
1/. Please provide a standard oxidation potential values for GSH and the investigated phenolics.
We have provided standard oxidation of GSH and closely related systems that contain catechols and pyrogallols. Please see the explanation provided in the revised version in lines 143 and below.
2/. Please provide a thermodynamic explanation for the regulating system based on these values of standard oxidation potentials.
Answer: The required explanation is now provided in lines 150 and below, taken from an experimental study of GSH and closely related catechins. The authors conclude that GSH is the worse reducing agent, and therefore, its oxidized form reduction by catechins (catechols and pyrogallols) is thermodynamically favored
3/. Regarding the living organisms, what is the concentration of phenolics and their conjugated forms in relation to the GSH in the physiological human fluids such as blood, respiratory tract lining fluid and cerebrospinal fluid?,
Answer: We expect that their concentration would be much lower than the GSH concentration, which is relatively large, up to 10 mM
4/. Please provide a data on the bioavailability of catechol (Cat), pyrogallol 96
(Pyr), resveratrol (Res) and piceatannol (Pic).
Answer: Since they are exogenous antioxidants, their concentrations depend on the diet and other variables. However, we have selected them as a kind of phenols with two or three OH groups, and it is widely known that such phenols are bioavailable and effective. Please see the explanation from line 118 of the revised version.
5/. Why these compounds were selected among more the twenty thousand secondary plant metabolites?
Answer: We have added in the introduction in lines 106 and bellow a discussion about the reasons for our selection; however, the main reason is that they are well-known kinds of antioxidants with excellent antioxidant activity.
6/. Does the regenerating system is active in relation to the conjugated forms of phenolics?
Answer: Yes, we have studied phenols' activity and their conjugated base activity considering pKas. It is now better explained in the current version. See the explanation from line 250.
Reviewer 2 Report
Review manuscript Antioxidants antioxidants-2380873 by Carrenon-Gonzales et al.
The manuscript by Carrenon-Gonzales et al. reports a computational study which discuss and rationalize a potential synergistic antioxidant behavior between thiols and phenolic antioxidants. It is expected that thiols will first work to counteract autoxidation thereby oxidizing to the thiyl radicals with are then reduced back by phenols. It is shown that the reaction of phenols with thyil radicals isvery fast both in water and in lipid media and occurs via the combination of a SET process and a PCET mechanism in water, while the PCET would be the only mechanism in lipids.
The manuscript is interesting and sound and it would certainly meet the interests of the journal’s readership, however some aspect need to be further improved before publication, as detailed in the following.
1) Page 2 lines 66-67. Authors present the regeneration of antioxidants such as phenols by superoxide as an hypothesis rising from previous computational prediction; indeed they apparently ignore that the regeneration of antioxidants by superoxide (or the neutral hydroperoxyl radicals) has been extensively demonstrated by experiment in recent work and the mechanism, the kinetics, the thermodynamics and some potential applications of their synergism have been discussed for two classes of antioxidants: nitroxides (see: J. Am. Chem. Soc. 2018, 140, 10354−10362) and phenols including polyphenols (Food Chemistry, 2021, 345, 128468; Angew. Chem. Int. Ed. 2021, 60, 15220–15224). Since those achievements strongly support the relevance of their own work, Authors should briefly address this matter in their introduction and include it in their discussion. In this conjunction, I recommend that a discussion section is added before the conclusions.
2) Page 3, line 102-104. Several examples of synergistic antioxidant behavior have been disclosed for chalcogen-containing phenols in the presence of cysteine derivatives, albeit the synergism was shown to occur in the opposite direction: thiols repaired the phenolic antioxidants (see for instance Chem. Eur. J. 2013, 19, 7510 – 7522; Biomolecules 2022, 12, 90. https://
doi.org/10.3390/biom12010090). Discussion on this opposite scenario faced of the one investigated in this study would significantly enhance the value of this work and I recommend considering it in the introduction or in the discussion section. In this regard dome discussion would greatly enhance the significance of the work as the synergic cycle is very briefly addressed just in the conclusion without providing much discussion on its relevance and biological consequence.
3) Page 3 schemes 1 and 2. Both schemes show the synergistic antioxidant cycle in aqueous and lipid medium (respectively) having the quenching of an alkyl radical by the thiol as the starting event of the cycle. This however can only have some relevance in completely anoxic conditions, since in the presence of oxygen, the diffusion-controlled reaction of alkyl radicals with O2 to form peroxyl radicals would kinetically outcompete the reaction of alkyl radicals with thiols. Since anoxic conditions are not representative of autoxidation settings (for obvious reasons) or of biological systems, authors should revise the schemes or better clarify, although such reaction is not the main object of their work.
4) Page 4 eq. 8: please check the subscript indexes.
5) Table 1: It is apparent that the calculated overall rate constants are largely driven by the SET process which is considered from phenolate anion to the thiyl radical to afford the thiolate (based on scheme 1). It is unclear how the pKa difference of anion reactant and product was used to account for the delta G° of the reaction. If I misunderstood and, actually, the reaction between the phenol and the thiyl radical was calculated also for SET in water please clarify providing a detailed scheme of the calculated reactions.
6) Tables 1 and 2. Where specific contribution of explicit solvent molecules considered, beside the PCM model? Ion-dipole interaction in water and H-bonding both in water and in and ester lipid model would substantially affect the kinetics. In the tables values for TS with and without H-bond are given. However, it is unclear which are the H-bonding partners and if solvent is considered.
This is further not explained in the subsequent text at lines 219-232. The higher rate constants calculated for H-bonded species, suggest that explicit solvent is not computed as H-bonding partner, which is conformed in figure 1. Please clarify this point in the text and tables and discuss the possible role of the solvent, as, if not explicitly included in the calculations, it would largely affect the kinetic results. I am not insisting that the authors extend the calculation to include explicit solvent molecules, however they should at least address the point in discussing their results.
Author Response
We thank the Reviewer for the comments that certainly improve the manuscript.
Comments:
1) Page 2 lines 66-67. Authors present the regeneration of antioxidants such as phenols by superoxide as an hypothesis rising from previous computational prediction; indeed they apparently ignore that the regeneration of antioxidants by superoxide (or the neutral hydroperoxyl radicals) has been extensively demonstrated by experiment in recent work and the mechanism, the kinetics, the thermodynamics and some potential applications of their synergism have been discussed for two classes of antioxidants: nitroxides (see: J. Am. Chem. Soc. 2018, 140, 10354−10362) and phenols including polyphenols (Food Chemistry, 2021, 345, 128468; Angew. Chem. Int. Ed. 2021, 60, 15220–15224). Since those achievements strongly support the relevance of their own work, Authors should briefly address this matter in their introduction and include it in their discussion. In this conjunction, I recommend that a discussion section is added before the conclusions.
Answer: Some of the suggested references have been included in the text (ref. 46 and 47). We thank you for showing us bibliography that experimentally supports the synergy between the superoxide radical and phenols. Although this addition enriches our work and helps to broaden the panorama, we believe that we should not detail the results shown in these references as it could divert the objective of our work. Please see the explanation provided in the revised version from line 67 and below.
2) Page 3, line 102-104. Several examples of synergistic antioxidant behavior have been disclosed for chalcogen-containing phenols in the presence of cysteine derivatives, albeit the synergism was shown to occur in the opposite direction: thiols repaired the phenolic antioxidants (see for instance Chem. Eur. J. 2013, 19, 7510 – 7522; Biomolecules 2022, 12, 90. https://
doi.org/10.3390/biom12010090). Discussion on this opposite scenario faced of the one investigated in this study would significantly enhance the value of this work and I recommend considering it in the introduction or in the discussion section. In this regard dome discussion would greatly enhance the significance of the work as the synergic cycle is very briefly addressed just in the conclusion without providing much discussion on its relevance and biological consequence.
Answer: As we mentioned, the synergy between thiols and phenols is known, but in the works already cited, the recommended references and the work we propose, the mechanisms, conditions and objectives change. We have included some of the recommended references since, although the approach between our work, that of the cited authors, and that of the recommended authors is different, it helps to enrich the study of the synergy between this class of antioxidants. See references 86 and 87. Please see the small explanation provided in the revised version from line 134.
3) Page 3 schemes 1 and 2. Both schemes show the synergistic antioxidant cycle in aqueous and lipid medium (respectively) having the quenching of an alkyl radical by the thiol as the starting event of the cycle. This however can only have some relevance in completely anoxic conditions, since in the presence of oxygen, the diffusion-controlled reaction of alkyl radicals with O2 to form peroxyl radicals would kinetically outcompete the reaction of alkyl radicals with thiols. Since anoxic conditions are not representative of autoxidation settings (for obvious reasons) or of biological systems, authors should revise the schemes or better clarify, although such reaction is not the main object of their work.
Answer: The scheme 1 has been modified to clarify the mechanism studied in aqueous medium. Although we know about the competing reaction (addition of O2 to alkyl radicals), we believe that given the concentrations of glutathione and O2 in the medium, the reaction between glutathione and alkyl radicals is favored. See the explanation from line 167 and below.
4) Page 4 eq. 8: please check the subscript indexes.
Answer: We have reviewed them, but we do not understand the error. It seems correct in our version.
5) Table 1: It is apparent that the calculated overall rate constants are largely driven by the SET process which is considered from phenolate anion to the thiyl radical to afford the thiolate (based on scheme 1). It is unclear how the pKa difference of anion reactant and product was used to account for the delta G° of the reaction. If I misunderstood and, actually, the reaction between the phenol and the thiyl radical was calculated also for SET in water please clarify providing a detailed scheme of the calculated reactions.
Answer: Although we have explained how the rate constants were calculated in the “materials and methods” section, we have added a brief explanation in the “results and discussion section”. Also, the scheme 1 has been modified to clarify the deprotonation step previous to electron transfer. Please see the explanation provided in the revised version from line 241 and below.
6) Tables 1 and 2. Where specific contribution of explicit solvent molecules considered, beside the PCM model? Ion-dipole interaction in water and H-bonding both in water and in and ester lipid model would substantially affect the kinetics. In the tables values for TS with and without H-bond are given. However, it is unclear which are the H-bonding partners and if solvent is considered.
This is further not explained in the subsequent text at lines 219-232. The higher rate constants calculated for H-bonded species, suggest that explicit solvent is not computed as H-bonding partner, which is conformed in figure 1. Please clarify this point in the text and tables and discuss the possible role of the solvent, as, if not explicitly included in the calculations, it would largely affect the kinetic results. I am not insisting that the authors extend the calculation to include explicit solvent molecules, however they should at least address the point in discussing their results.
Answer: We do not use the PCM solvation model. We use the SMD solvation model (ref.95, "material and methods" section, line 184 in the revised version). Although the SMD model is also compatible with the use of explicit solvent molecules, based on our experience and work in the area, we know that for large systems involving transition states, making use of explicit solvent molecules can become an impossible job. Find the number of explicit solvent molecules needed to stabilize the system, the different geometries that the system can adopt, the number of solvation spheres, etc. it is a job that requires a lot of time and it is not our main objective and it is not necessary except for description of small ions for example when describing SET to OOH or OH radical the error of SMD (or any other continuum model) is about 7 kcal/mol for phenoxyl radical it drops below 1 kcal/mol. Please see the explanation provided in the revised version from line 260 and below.
Reviewer 3 Report

Moderate editing of English language is required.
Author Response
We thank the Reviewer for the comments that certainly improve the manuscript.
Comments:
The paper by Carreno-Gonzalez and Alvarez-Idaboy describes the quantum chemicstry calculations for the reaction of glutathionyl radical (GS• ) with several antioxidants to regenerate glutathione (GSH) in aqueous and lipid media. There are several minor points as indicated below that need to be addressed by the authors before publication in the journal.
- “Theoretical” or “Using Quantum Chemistry Calculations” should be inserted in the title, since this paper does not deal with experimental results and discussions.
Answer: We have added “: A theoretical insight.” To the title
- The rate constants should be described, for example, to be 3.21 ´ 106 M–1 s–1 rather than 3.21E+06 M-1 s-1 . A space is required between “M–1 ” and “s–1 .”
Answer: It has been corrected
- “O” in “O2 •– ” is not in italic.
Answer: It is now in italics
- Scheme 1 does not show the PCET mechanism but the SPLET mechanism. 5. If the regeneration mechanism of GSH includes the electron transfer from phenolate anions to GS• , the deprotonation process of antioxidants should be taken into account.
Answer: We have changed the mechanism to SPLET in all the manuscript using SET only for the second step of the SPLET process.
Reviewer 4 Report
In this work the possible molecular mechanisms between glutathione and some phenolic antioxidants such as catechol, pyrogallol, resveratrol, and piceatannol were investigated using computational techniques. The main hypothesis is that the thiol-type antioxidants, among them glutathione radical (GS•) may be regenerated by phenolic-type antioxidants. The authors estimated the possibility of the main two different type of mechanisms, the single-electron transfer (SET) and the proton-coupled electron transfer (PCET) in aqueous and lipid media. The performed analyses related to the earlier finding that the superoxide anion radical can repair oxidized DNA through SET mechanism, and the authors in this paper showed that it also could restore phenols. In my opinion, the thermodynamic, kinetics evaluation of the mechanisms via glutathione and phenols might be part of a synergistic cycle has a great importance. I miss experimental evidence for the results, but I am sure that these results can be basis of different experimental approaches in the future. Unfortunately, I am not familiar either with the quantum chemistry or computational kinetics. I feel that - because of the methods used in this manuscript - this paper may fit into the Oxidative Stress and Antioxidants in Computational Chemistry Special Issue of the Antioxidants.
Author Response
We appreciate the positive comments of the Reviewer and his/her sincerity regarding Quantum Chemistry and Computational Kinetics. The opinion of an expert in experiments is more important for us since we have no doubts about the methodology used.